# Distributed Ellipsoidal Intersection Fusion Estimation for Multi-Sensor Complex Systems

**DOI:** 10.3390/s22114306

**Published:** 2022-06-06

**Authors:** Peng Zhang, Shuyu Zhou, Peng Liu, Mengwei Li

**Affiliations:** 1School of Instrumentation and Electronic, North University of China, Taiyuan 030051, China; zhangpeng6@nuc.edu.cn (P.Z.); shuyuz0809@163.com (S.Z.); lmw@nuc.edu.cn (M.L.); 2Academy for Advanced Interdisciplinary Research, North University of China, Taiyuan 030051, China; 3North Automatic Control Technology Institute, Taiyuan 030006, China

**Keywords:** data fusion, unknown input interference, measure propagation delay, unknown correlation

## Abstract

This paper investigates the problem of distributed ellipsoidal intersection (DEI) fusion estimation for linear time-varying multi-sensor complex systems with unknown input disturbances and measurement data transmission delays. For the problem with external unknown input disturbance signals, a non-informative prior distribution is used to model the problem. A set of independent random variables obeying Bernoulli distribution is also used to describe the situation of measurement data transmission delay caused by network channel congestion, and appropriate buffer areas are added at the link nodes to retrieve the delayed transmission data values. For multi-sensor systems with complex situations, a minimum mean square error (MMSE) local estimator is designed in a Bayesian framework based on the maximum a posteriori (MAP) estimation criterion. In order to deal with the unknown correlations among the local estimators and to select the fusion estimator with lower computational complexity, the fusion estimator is designed using ellipsoidal intersection (EI) fusion technique, and the consistency of the estimator is demonstrated. In this paper, the difference between DEI fusion and distributed covariance intersection (DCI) fusion and centralized fusion estimation is analyzed by a numerical example, and the superiority of the DEI fusion method is demonstrated.

## 1. Introduction

In recent years, multi-sensor systems have been widely used in sensor networks, artificial intelligence, combinatorial navigation, and industrial control. Since multi-sensor systems can provide more information for more accurate control of the system, it makes the information fusion estimation techniques of multi-sensor systems receive wide attention and have important research significance [1,2,3,4,5,6,7]. In complex systems with multiple sensors, the methods of information fusion estimation are generally divided into centralized fusion estimation and distributed fusion estimation. The principle is to fuse multiple estimates into one highly reliable estimation method according to the corresponding fusion algorithm [8]. In centralized fusion estimation, the measurement data from multiple sensors are processed by using state measurement enhancement methods. In contrast, distributed fusion estimation, with its unique parallel structure, puts the local state estimates of different sensors into the fusion center and follows the corresponding fusion rules for state estimation [9].

The centralized fusion estimator can provide the best estimation accuracy when all sensors are working properly. However, if the sensors fail in operation, the centralized fusion estimator cannot detect and discard the faulty sensors in time, leading to a decrease in the reliability of the fusion estimation results and an increase in the error. A suboptimal distributed estimator with a parallel structure can solve this problem well. The presence of a parallel structure makes it easy to detect and isolate the faulty sensors, so the distributed estimator has good reliability and flexibility [10,11,12]. At the same time, in centralized fusion estimators, the system incurs expensive computational costs as the number of sensors continues to increase. Compared to centralized fusion estimation, distributed fusion estimation has a much lower computational cost [13]. In multi-sensor complex systems, the choice of an estimation algorithm with high accuracy and low computational cost is crucial in the face of computational resource limitations and uncertainty in the occurrence of system failures. Therefore, the use of a distributed fusion estimator is one of the motivations of this paper.

### 1.1. Related Work

In a multi-sensor complex system, the system is affected by some network-induced phenomena due to the uncertainty of the network heterogeneous model and the occurrence of sensor failures. For example, unknown external information disturbances, random delays in measurement data, and packet loss. For systems with unknown inputs or disturbances, these disturbances may be invariant, time-varying, or random [14,15]. In [16], the problem of state estimation for systems subjected to unknown input disturbances during sensor measurements is presented, an optimal state estimator is designed, and the results are applied to generalized systems with unknown inputs [17,18]. In [17], good results were obtained by using unbiased minimum variance (UMV) estimation for systems with unknown inputs. Unlike other methods, in [19], the unknown input information is modeled using a non-informative prior distribution, and a minimum mean square error (MMSE) estimator is designed to estimate the system in a Bayesian framework.

Meanwhile, network congestion occurs due to limited communication bandwidths in sensor networks. Random delays are inevitable when transmitting measurement data. The delay phenomenon is inevitably accompanied by packet loss, which significantly affects the performance of the network system [20]. When dealing with random delays in the transmission of measurement data, a set of independent Bernoulli-distributed random variables or a Markov chain can be used to describe the transmission random delay phenomenon. In [21], the optimal filtering problem for systems with Markov chain communication delays is studied. Meanwhile, in [16], a set of Bernoulli-distributed random variables is introduced to describe the stochastic delay phenomenon. In order to avoid packet loss as much as possible, in [19], delay measurements are retrieved by introducing a finite length buffer at the link nodes. In systems with delay phenomena, both measurement enhancement techniques and replication retransmission can make good use of measurement delay data [22,23].

Facing the problem of computational resource limitations and system uncertainty in complex systems with multiple sensors, a distributed fusion estimator is used to estimate the system. Despite the rapid rise of the distributed fusion estimation in recent years, it is often plagued by unknown correlation information in sensor networks, which prevents the design of fusion estimators with high accuracy [24]. Currently, the main methods that can solve fusion estimation with unknown correlations are: covariance intersection (CI) fusion methods and ellipsoidal intersection (EI) fusion methods. In [25], the CI fusion method was proposed. It parameterizes the fusion estimation by converting it into a convex combination problem of two local estimates. Once the idea was proposed, it inspired many people in the field to pursue it. Despite some improvements to the CI fusion method, the accuracy of the fusion results still shows a decreasing trend. The reason for this decline is that the choice of CI fusion parameterization is a fusion formula that directly bypasses the discussion of the relevance of the local estimates and yields fusion results that are too conservative [26]. To pursue higher accuracy to accurately control the system. In [27], the EI fusion method was proposed to redefine a fusion parameterization. It expresses the correlation between the local estimates in an algebraic formulation through the parameterization before deriving the fusion estimates based on the conditions of the local estimates, and the algebraic fusion formulation ensures that the EI fusion algorithm reduces the computational complexity. In contrast, the EI fusion algorithm solves the difficult problem of unknown correlations between local estimates more effectively [28,29].

### 1.2. Paper Contributions

In this paper, we study the problem of data fusion estimation for a linear time-varying multi-sensor complex system with two network-induced phenomena of both unknown input disturbances and measurement transmission delays. In order to obtain a fusion estimator with high accuracy and low computational cost, distributed fusion estimation is used in this paper for the system estimation. In this case, the information of unknown input perturbations is modeled by a non-informative prior distribution. All possible values are described by using a probability density function. The randomness of the measured data transmission delay is described by a set of independent random variables obeying Bernoulli distribution, and a buffer of finite length is added at the link node to obtain the data set for the delay measurement. For the design of the system local estimator, the MMSE local estimator is designed in a Bayesian framework based on the nature of the state-conditional distribution and the maximum a posteriori (MAP) estimation criterion. When fusion processing is performed on the local estimates, the correlation between the local estimates is unknown due to the randomness of the measurement data delay, which makes it difficult to obtain fusion results with high accuracy. To solve the problem of fusion estimation with unknown correlations between local estimates, a distributed ellipsoidal intersection (DEI) fusion estimator is designed by analyzing the distributed fusion algorithms. Compared with the distributed covariance intersection (DCI) fusion estimation, the problem of overly conservative estimation results is solved, and the estimation accuracy is improved. The parallel structure makes the designed estimator less computationally expensive and reduces the computational complexity than the centralized fusion estimator.

### 1.3. Paper Outline

The structure of this paper is as follows. In Section 2, we describe two network-induced phenomena in multi-sensor complex systems: unknown input interference and measurement data transmission delay. Section 3 designs a local estimator for multi-sensor complex systems based on the MMSE criterion. Section 4 determines the estimation of the multi-sensor complex system using the DEI fusion estimator and shows the consistency of the designed DEI fusion estimator. The numerical simulation results and computational complexity analysis are given in Section 5. The conclusions are given in Section 6.

## 2. Problem Description

Let us consider a multi-sensor linear time-varying system disturbed by unknown input information:(1)$$x_{k+1}=A_{k}x_{k}+D_{k}d_{k}+\omega_{k}$$
where $x_{k}\in R^{n}$ denotes the state estimation vector at moment *k*, $A_{k}$ denotes the state matrix that is time-varying and matches the dimensionality of $x_{k}$, $d_{k}\in R^{p}$ denotes the external input vector, $D_{k}$ denotes the time-varying matrix that matches the dimensionality of $d_{k}$, and the process noise is described by $\omega_{k}\in R^{n}$, which has a mean of 0 and covariance matrix $\mathrm{of}\;Q_{k}>0$. Additionally, we give the measurement equations for the sensors in the system that measure the data:(2)$$y_{i,k}=C_{i,k}x_{k}+v_{i,k},\;i=1,\ldots ,L$$
where $y_{i,k}\in R^{m_{i}}$ denotes the measured data values in the *i*th network transmission channel with the total number of sensors *L*. $C_{i,k}$ denotes the time-varying matrix matching the dimensionality of $x_{k}$, and $v_{i,k}\in R^{m_{i}}$ denotes the measurement noise with a mean of 0 and covariance of $R_{i,k}>0$. The measurement noise of each measurement channel is independent of each other, and the initial state $x_{0}$, which obeys a Gaussian distribution, is also uncorrelated with $\omega_{k}$ and $v_{i,k}$.

For the problem of data fusion estimation of a multi-sensor linear time-varying system with unknown input disturbances and measurement data transmission delays, the flow structure of the system is shown in Figure 1. The system works as follows: first, the multi-sensor system subject to unknown external input disturbances is measured by a multi-sensor to obtain information about the system state at each moment. The obtained measurement information is transmitted to the corresponding link nodes through the network channel, and a series of local state estimates are generated in the designed MMSE estimator. The local state estimates are fused at the fusion center to obtain the estimation results.

Since the external input vector $d_{k}$ is unknown and its information is not available, it cannot participate in the design of the estimator. In order not to affect the design of the estimator, it is guaranteed that the estimation accuracy will not be biased. In [19], an assumption is adopted: the number of channels of external input disturbance is guaranteed to be smaller than the number of channels of state estimation by controlling the rank of the time-varying matrix $D_{k}$: $\mathrm{rank}\left(D_{k}\right)=p,p<n$. The proposed assumption is guaranteed by this intuitive formulation. Additionally, since all possible values of the unknown input vector $d_{k}$ appear with equal probability, we model $d_{k}$ using a non-informative prior distribution [16]. The probability density function of f is, i.e.,
(3)$$f\left(d_{k}\right)\propto 1$$

Inspired by [16,19], for our proposed hypothesis, the matrix of unknown input coefficients $D_{k}$ should strictly adhere to the matrix column full rank to ensure that the number of channels of the external input disturbances is smaller than the number of channels of the state estimation. Meanwhile, a new matrix $D_{k}^{⊥}$ is constructed under the principle of orthogonal complementation, such that matrix $D_{k}^{⊥}$ satisfies the rules of $\left[D_{k}\;D_{k}^{⊥}\right]\in R^{n\times n}$, $\mathrm{rank}\left[D_{k}\;D_{k}^{⊥}\right]=n$, and $D_{k}^{T}D_{k}^{⊥}=0$.

When the measurement information is transmitted through the network channel, the measurement delay occurs randomly, because the limited bandwidth of the network channel causes the channel congestion phenomenon. During transmission, if the measurement information is not received by the link node within a given time interval, packet loss occurs, so the packet loss phenomenon also exists at the same time [20]. Since the measurement delay and packet loss occur randomly, we adopt the Bernoulli distribution random variable approach to describe the phenomena triggering the measurement delay and packet loss [16]. First, we assume that the measurement data $y_{i,k}$ is a delay in the network channel for $\theta_{i,k}$ moments and $\theta_{i,k}$ is a random variable. We model the random variable $\theta_{i,k}$ by using the probability mass function $f_{i}$:(4)$$f_{i}\left(j\right)=\mathrm{Pr}\left[\theta_{i,k}=j\right],i=1,2,\cdots ,L,\;\;j=0,1,2,\cdots$$
where $\theta_{i,k}$ for different channels and different moments are independent of each other. If no buffer exists at the link node, the transmission of measurement data from the sensor to the link node is considered successful only if $\theta_{i,k}=0$; otherwise, the transmission fails. Therefore, the process of measurement data $y_{i,k}$ from the sensor to the link node is considered as a Bernoulli process. To solve the problem of the randomness of $\theta_{i,k}$, we obtain the information of the random variable $\theta_{i,k}$ by adding an appropriate buffer at the link node and by measuring the time of data reacquisition from the buffer [19]. Here, we assume a buffer of length $\varepsilon_{i}\left(\varepsilon_{i}\geq 2\right),$ so that the link node can receive all measurement data with a delay time of $k-\varepsilon_{i}+1$. The earliest measurement update value $\kappa_{i,k}$ for the *k*th moment and the *i*th buffer is defined as:(5)$$\kappa_{i,k}=\left\{\begin{array}{c} t,0<k-\varepsilon_{i}+1\leq t<k\;\\ k,t\geq k \end{array}\right.$$

The receipt of the measurement $y_{i,k}$ is indicated by introducing a sequence of binary variables $\gamma_{t,k}^{i}$. When $\gamma_{t,k}^{i}=1$, it indicates that the measurement is received at the *k*th moment or before. When the delay time is equal to or greater than $\varepsilon_{i}$, the measurement data will be discarded, and this case is considered as a packet loss phenomenon. We define the set of measurements in the *i*th buffer at the *k*th moment by defining $\ell_{i,k}$, i.e.,
(6)$$\ell_{i,k}≜\left\{\left(\gamma_{i,0}y_{i,0}\right),\left(\gamma_{i,1}y_{i,1}\right),\cdots ,\left(\gamma_{i,t}y_{i,t}\right),\cdots ,\left(\gamma_{i,k}y_{i,k}\right)\right\}$$
where $\gamma_{i,t}=\gamma_{t,k}^{i},$ and our goal is to obtain an estimation problem for the state $x_{k}$ conditional on the set of measurements $\ell_{k}≜\left\{\ell_{1,k},\ell_{2,k},\cdots ,\ell_{L,k}\right\}$.

## 3. Local Estimation of Complex Multi-Sensor Systems

In this section, in order to solve the problem of estimating the state $x_{k}$ conditional on the measurement set $\ell_{k}$, we need to design local estimators at each link node to obtain the state estimates. Usually, the estimation for the state is often based on one observation, and the estimator is often designed in a Bayesian framework. Since the state $x_{k}$ is estimated based on the measurement set $\ell_{k}$, the unknown input $d_{k}$ is modeled using a non-informative prior distribution. According to the standard results of optimal estimation, the MMSE estimate is equivalent to the mean of the state-conditional distribution conditional on the measurement set, so the design of the local estimator can be performed using the MMSE estimation approach. Our goal in designing the local estimator is to find the recursive problem of the conditional distribution of the state $x_{k}$ conditional on the measurement set $\ell_{k}$.

To ensure that the local estimator design is error-free, we have to verify that the coefficient matrix $\mathrm{rank}\left(D_{k}\right)=p$ of the unknown input interference signal satisfies the assumption that p<n. By introducing $T_{k}≜{\left[D_{k}\;D_{k}^{⊥}\right]}^{-1}$ and $L_{k}≜\left[0\;I_{n-p}\right]T_{k}$, then using ${\left[C_{i,k}^{T}\;L_{k-1}^{T}\right]}^{T}\in R^{\left[m+\left(n-p\right)\right]\times n}$ to obtain m≥p, which leads to $\mathrm{rank}{\left[C_{i,k}^{T}\;L_{k-1}^{T}\right]}^{T}=n$, this verifies the hypothesis that the number of independent measurement channels is not less than the number of channels of the unknown external input by the rank of the coefficient matrix $D_{k}$ [19]. The above hypothesis will automatically hold when the measurement matrix $C_{i,k}$ satisfies the condition of full column rank. For the system that satisfies the stated assumptions, we can verify the rank of matrix $D_{k}$ by expressions based on the system expressions, regardless of whether the system is time-varying or not, ensuring the accuracy of the estimation results.

For systems that satisfy the condition of $\mathrm{rank}\left(D_{k}\right)=p$, we base the design of the estimator of state $x_{k}$ on the condition of the measurement set $\ell_{k}$ by representing in a Bayesian framework, i.e.,
(7)$$P_{X|\ell }\left(x_{k}|\ell_{k}\right)=\frac{P_{\ell |X}\left(\ell_{k}|x_{k}\right)p_{X}\left(x_{k}\right)}{P_{\ell }\left(\ell_{k}\right)}$$
where $P_{\ell |X}\left(\ell_{k}|x_{k}\right)$ denotes the likelihood probability distribution and $p_{X}\left(x_{k}\right)$ denotes the prior probability distribution.

The set of measurements $\ell_{i,k}≜\left\{\left(\gamma_{i,0}y_{i,0}\right),\left(\gamma_{i,1}y_{i,1}\right),\cdots ,\left(\gamma_{i,t}y_{i,t}\right),\cdots ,\left(\gamma_{i,k}y_{i,k}\right)\right\}$ in the buffer of the *i*th link node, the posterior probability distribution of the state $x_{k}$ conditional on $\ell_{i,k}$ is:(8)$$P_{X|\ell }\left(x_{k}|\gamma_{i,k}y_{i,k}\right)=\frac{P_{\ell |X}\left(\gamma_{i,k}y_{i,k}|x_{k}\right)p_{X}\left(x_{k}\right)}{P_{\ell }\left(\gamma_{i,k}y_{i,k}\right)}$$

The prior probability distribution $p_{X}\left(x_{k}\right)=P_{X|\ell }\left(x_{k}|\ell_{i,k-1}\right)$, due to the non-informative prior distribution modeling the unknown input disturbance information $d_{k}$, is obtained according to the full probability formula:(9)$$P_{X|\ell }\left(x_{k}|\ell_{i,k-1}\right)=\overset{}{\underset{R^{P}}{\int }}P_{X|\ell }\left(x_{k}|\ell_{i,k-1},d_{k-1}\right)P(d_{k-1}|\ell_{i,k-1})d\;d_{k-1}$$

Converting Equation (9) to the Gaussian distribution form yields
(10)$$P_{X|\ell }\left(x_{k}|\ell_{i,k-1}\right)\propto \int_{R^{P}}^{}\exp\left[-\frac{1}{2}{\left(x_{k}-{\hat{x}}_{i,k|k-1}\right)}^{T}{\left({\hat{P}}_{i,k|k-1}\right)}^{-1}\left(x_{k}-{\hat{x}}_{i,k|k-1}\right)\right]dd_{k-1}$$
where ${\hat{x}}_{i,k|k-1}=A_{k-1}{\hat{x}}_{i,k-1}+D_{k-1}d_{k-1}$ and the error covariance is ${\hat{P}}_{i,k|k-1}=A_{k-1}{\hat{P}}_{i,k-1}A_{k-1}^{T}+Q_{k-1}$.

According to the nature of the marginal distribution of the multivariate Gaussian distribution, Equation (10) is organized to obtain:(11)$$P_{X|\ell }\left(x_{k}|\ell_{i,k-1}\right)\propto \exp\left[-\frac{1}{2}{\left(x_{k}-A_{k-1}{\hat{x}}_{i,k-1}\right)}^{T}L_{k-1}^{T}{\left(L_{k-1}{\hat{P}}_{i,k|k-1}L_{k-1}^{T}\right)}^{-1}L_{k-1}\left(x_{k}-A_{k-1}{\hat{x}}_{i,k-1}\right)\right]$$

Based on Equation (11), it is known that the prior probability $p_{X}\left(x_{k}\right)$ obeys a Gaussian distribution, i.e.,
(12)$$p_{X}\left(x_{k}\right)=N\left({\bar{x}}_{i,k},{\bar{P}}_{i,k}\right)$$
where ${\bar{x}}_{i,k}=A_{k-1}{\hat{x}}_{i,k-1}$ and ${\bar{P}}_{i,k}={\left[L_{k-1}^{T}{\left(L_{k-1}{\hat{P}}_{i,k|k-1}L_{k-1}^{T}\right)}^{-1}L_{k-1}\right]}^{-1}$.

Under the condition that the prior probability distribution $p_{X}\left(x_{k}\right)$ follows a Gaussian distribution and the measurement noise also follows a Gaussian distribution, the MMSE estimate is equivalent to the MAP estimate, so it can be converted to find the MAP estimate. The posterior probability distribution is proportional to the product of the likelihood probability and the prior probability, and since the prior distribution has been found, the likelihood probability distribution $P_{\ell |X}\left(\gamma_{i,k}y_{i,k}|x_{k}\right)$ is calculated.
(13)$$P_{\ell |X}\left(\gamma_{i,k}y_{i,k}|x_{k}\right)\propto exp\left[-\frac{1}{2}{\left(\gamma_{i,k}y_{i,k}-\gamma_{i,k}C_{i,k}x_{k}\right)}^{T}\gamma_{i,k}R_{i,k}^{-1}\left(\gamma_{i,k}y_{i,k}-\gamma_{i,k}C_{i,k}x_{k}\right)\right]$$

Based on the measurement set $\ell_{i,k}$, the posterior probability distribution of the state $x_{k}$ is:(14)$$P_{X|\ell }\left(x_{k}|\gamma_{i,k}y_{i,k}\right)\propto P_{\ell |X}\left(\gamma_{i,k}y_{i,k}|x_{k}\right)p_{X}\left(x_{k}\right)$$

The maximized posterior probability distribution function is:(15)$${\hat{x}}^{MAP}\left(\gamma_{i,k}y_{i,k}\right)=argmax\;P_{X|\ell }\left(x_{k}|\gamma_{i,k}y_{i,k}\right)p_{X}\left(x_{k}\right)$$
where ${\hat{x}}^{MAP}\left(\gamma_{i,k}y_{i,k}\right)$ is called the maximum a posteriori estimator of $x_{k}$.

Substituting Equations (11) and (13) into (14), we obtain the posterior probability distribution $P_{X|\ell }\left(x_{k}|\gamma_{i,k}y_{i,k}\right),$ satisfying the Gaussian distribution of the form:(16)$$P_{X|\ell }\left(x_{k}|\gamma_{i,k}y_{i,k}\right)\propto exp\left[-\frac{1}{2}{\left(x_{k}-\mu_{i,k}\right)}^{T}\Pi_{i,k}^{-1}\left(x_{k}-\mu_{i,k}\right)\right]$$
where ${\hat{x}}^{MAP}=\mu_{i,k}=A_{k-1}{\hat{x}}_{i,k-1}+\gamma_{i,k}\Pi_{i,k}(C_{i,k}^{T}y_{i,k}R_{i,k}^{-1}-C_{i,k}^{T}R_{i,k}^{-1}C_{i,k}A_{k-1}{\hat{x}}_{i,k-1})$, the covariance matrix is $\Pi_{i,k}={\left[\gamma_{i,k}C_{i,k}^{T}R_{i,k}^{-1}C_{i,k}+L_{k-1}^{T}{\left(L_{k-1}{\hat{P}}_{i,k|k-1}L_{k-1}^{T}\right)}^{-1}L_{k-1}\right]}^{-1}$.

Since the prior probability distribution and the measurement noise obey Gaussian distribution, i.e.,
(17)$${\hat{x}}^{MMSE}=E\left[x_{k}|\gamma_{i,k}y_{i,k}\right]={\hat{x}}^{MAP}$$

Thus, we obtain a local estimator of the Gaussian distribution of state $x_{k}$ for a time-varying linear multi-sensor complex system with unknown input disturbances and measurement data transmission delays under the condition of a measurement set $\ell_{i,k,}$ satisfying the condition of $\mathrm{rank}\left(D_{k}\right)=p$:(18)$${\hat{x}}_{i,k}=A_{k-1}{\hat{x}}_{i,k-1}+\gamma_{i,k}{\hat{P}}_{i,k}(C_{i,k}^{T}y_{i,k}R_{i,k}^{-1}-C_{i,k}^{T}R_{i,k}^{-1}C_{i,k}A_{k-1}{\hat{x}}_{i,k-1})$$
(19)$${\hat{P}}_{i,k}={\left[\gamma_{i,k}C_{i,k}^{T}R_{i,k}^{-1}C_{i,k}+L_{k-1}^{T}{\left(L_{k-1}{\hat{P}}_{i,k|k-1}L_{k-1}^{T}\right)}^{-1}L_{k-1}\right]}^{-1}$$

Our goal is to fuse the obtained local estimates at the fusion center to obtain estimation results with high accuracy.

## 4. Distributed Ellipsoidal Intersection (DEI) Fusion Estimation for Multi-Sensor Complex Systems

In this section, in order to solve the fusion problem of multi-sensor local estimation, a distributed fusion estimation algorithm suitable for linear multi-sensor time-varying discrete systems with unknown input disturbances and measurement transmission delays is selected. When we fuse the local estimates, we first consider the optimal matrix-weighted distributed fusion method for fusion estimation with the following fusion equation:(20)$${\hat{x}}_{k}=\overset{L}{\underset{i=1}{\sum }}\Omega_{i,k}{\hat{x}}_{i,k},\;\;i=1,\ldots ,L$$
where $\Omega_{i,k}$ denotes the optimal weight matrix and $\overset{L}{\underset{i=1}{\sum }}\Omega_{i,k}=I$. However, since the optimal weight matrix depends on the information of the mutual covariance ${\hat{P}}_{k}^{i,j}\left(i\neq j\right)$ between multi-sensors, and the proposed multi-sensor system is the phenomenon of a measurement transmission delay, the delay variables in the channel are all randomly occurring, resulting in a correlation between sensors that cannot be obtained [24]. An unknown correlation means that the mutual covariance $\mathrm{cov}\left(x_{i},x_{j}\right)$ is not computable, so it is difficult for us to obtain the analytic expression of the mutual covariance ${\hat{P}}_{k}^{i,j}$ between sensors, which causes some difficulties in the design of the fusion estimator.

Currently, a commonly used method in dealing with fusion estimation of unknown correlations is the CI fusion technique, which parameterizes the fusion formula and avoids the determination of the expression for the mutual correlation covariance $\mathrm{cov}\left(x_{i},x_{j}\right)$ [25]. Although this approach is generally accepted, the CI fusion approach is suboptimal. Since the CI fusion technique focuses on the analysis of the fusion formula rather than the correlation, it may lead to conservative results of a fusion [26]. Based on this situation, there is another method that parameterizes the local estimates when dealing with the case of unknown correlations: the EI fusion method. This parametric approach introduces three new estimates that provide an explicit description of the correlation and expresses the information about the correlation in an explicit expression. Both conservative estimations are avoided, while the extraction of unknown correlation information is taken into account, and the accuracy of the fusion is guaranteed [27].

Next, we analyze the EI fusion method. First, we consider two random vectors: $x_{i}\;\mathrm{and}\;x_{j}\in R^{n}$ with Gaussian distribution characteristics, which are both two prior estimates of the state vector $x\in R^{n}$, i.e.,
$$x_{i}∼N\left({\hat{x}}_{i},P_{i}\right),x_{j}∼N\left({\hat{x}}_{j},P_{j}\right)$$

Our goal is to fuse the two prior estimates into a new estimate $x_{f}$ that also obeys a Gaussian distribution, i.e.,
$$x_{f}∼N\left({\hat{x}}_{f},P_{f}\right)$$

It is also important to ensure that the fusion results of these two prior estimates satisfy the consistency of the fusion estimates, i.e., $P_{f}≼P_{i}$ and $P_{f}≼P_{j}$.

To characterize the unknown correlation, the EI fusion technique is performed by introducing three new two–two independent random vectors $x_{ii},x_{ij},x_{jj}\in R^{n}$ with a mean of $\mu_{ii},\gamma ,\mu_{jj}\in R^{n}$ and variance of $\Phi_{ii},\;\Gamma ,\;\Phi_{jj}\in R^{n\times n}$, respectively. The priori estimates $x_{i},x_{j}$ are represented by the information of $x_{ii},x_{ij},x_{jj}$ by constructing a new function Ψ:(21)$$x_{i}:=\Psi \left(x_{ii},x_{ij}\right)={\left(\Phi_{ii}^{-1}+\Gamma^{-1}\right)}^{-1}\left(\Phi_{ii}^{-1}\mu_{ii}+\Gamma^{-1}\gamma \right)\;x_{j}:=\Psi \left(x_{jj},x_{ij}\right)={\left(\Phi_{jj}^{-1}+\Gamma^{-1}\right)}^{-1}\left(\Phi_{jj}^{-1}\mu_{jj}+\Gamma^{-1}\gamma \right)$$
where ${\left(\Phi_{ii}^{-1}+\Gamma^{-1}\right)}^{-1}$ is denoted as the variance $P_{i}$ of the priori estimate $x_{i}$ and ${\left(\Phi_{ii}^{-1}+\Gamma^{-1}\right)}^{-1}\left(\Phi_{ii}^{-1}\mu_{ii}+\Gamma^{-1}\gamma \right)$ is the mean ${\hat{x}}_{i}$.

According to the relationship between the random vectors $x_{ii},x_{ij},x_{jj}$ and the priori estimates $x_{i},x_{j}$, we can express the correlation covariance $\mathrm{cov}\left(x_{i},x_{j}\right)$ of the priori estimates $x_{i},x_{j}$, i.e.,
(22)$$\mathrm{cov}\left(x_{i},x_{j}\right):=E{\left[x_{i}x_{j}\right]}^{T}-E\left[x_{i}\right]E{\left[x_{j}\right]}^{T}=P_{i}\Gamma^{-1}P_{j}$$

Since the correlation is unknown, to obtain a description of an arbitrary correlation, the information of the mutual correlation covariance $\mathrm{cov}\left(x_{i},x_{j}\right)$ is maximized. Based on the determinant of the mutual correlation covariance, it follows from Equation (22) that the problem of maximizing $\mathrm{cov}\left(x_{i},x_{j}\right)$ can be transformed into the problem of minimizing Γ, i.e.,
(23)$$\Gamma :=\arg\min\log\left|T\right|\;\mathrm{subject}\;\mathrm{to}\;T≽P_{i},T≽P_{j}$$

For random vectors $N\left(\hat{x},P\right),$ obeying Gaussian distribution can all be represented by the sublevel set ${Ɛ}_{\hat{x},P}=\{x\in R^{n}|{\left(x-\hat{x}\right)}^{T}P^{-1}\left(x-\hat{x}\right)\leq 1\}$. To represent minimal Γ intuitively, minimally Γ is characterized as the minimal ellipse containing ${Ɛ}_{{\hat{x}}_{i},P_{i}}\cup {Ɛ}_{{\hat{x}}_{j},P_{j}}$.

Since the prior estimates can be described by the introduced random variables, the fusion of the prior estimates is equivalent to the fusion of the introduced random variables, and by conditioning the function of Ψ, the fusion results are presented, i.e.,
(24)$$x_{f}:=\Psi \left(x_{i},x_{j}\right)=\Psi \left(\Psi \left(x_{ii},x_{ij}\right),x_{jj}\right)$$

Substituting Equation (21) and the variable information into Equation (24), it is obtained that
(25)$$P_{f}={\left(P_{i}^{-1}+P_{j}^{-1}-\Gamma^{-1}\right)}^{-1}\;{\hat{x}}_{f}=P_{f}\left(P_{i}^{-1}{\hat{x}}_{i}+P_{j}^{-1}{\hat{x}}_{j}-\Gamma^{-1}\gamma \right)$$

In order to pursue a computationally inexpensive fusion algorithm, the mean γ and variance Γ of the random variable $x_{ij}$ are represented with the information of a priori estimate [27], i.e.,
(26)$$\Gamma =S_{i}D_{i}^{\frac{1}{2}}S_{j}D_{\Gamma }S_{j}^{-1}D_{i}^{\frac{1}{2}}S_{i}^{-1}\;\gamma ={\left(P_{i}^{-1}+P_{j}^{-1}-2\Gamma^{-1}+2\eta I\right)}^{-1}\times \left((P_{j}^{-1}-\Gamma^{-1}+\eta I){\hat{x}}_{i}+\left(P_{i}^{-1}-\Gamma^{-1}+\eta I\right){\hat{x}}_{j}\right)$$
where ${\left[D_{\Gamma }\right]}_{qq}=\max\left\{1,{\left[D_{j}\right]}_{qq}\right\},q=1,\cdots ,n$. The eigenvalue decomposition $P_{i}=S_{i}D_{i}S_{i}^{-1}$ of the matrix $P_{i}$ yields the eigenvector matrix $S_{i}$ and the eigendiagonal matrix $D_{i}$. The positive definite matrix can be the square root decomposed as $A=LL^{T}$. According to the transformation relation in [28], it is obtained that $D_{i}^{-\frac{1}{2}}S_{i}^{-1}P_{j}S_{i}D_{i}^{-\frac{1}{2}}=S_{j}D_{j}S_{j}^{-1}$. Since the minimization Γ is the shape of the minimum ellipsoid of $P_{i}$ and $P_{j}$, $D_{\Gamma }=\max\left\{1,D_{j}\right\}$. This gives an algebraic expression for the correlation information between the local estimates.

Based on the above description, it can be seen that the EI fusion technique provides both an explicit description of the unknown correlation between the priori estimates and a parameterization of the fusion formula to ensure the accuracy and computational cost of the fusion results.

Since the obtained local estimates obey a Gaussian distribution, we use the EI fusion algorithm to design the estimator. In order to reduce the computational complexity of the fusion estimator, we use the method of fusing the local estimates in two by following the sequential fusion and obtain the final fusion estimate by performing the EI fusion process *L* – 1 times [15]. The distributed sequential EI fusion estimator is as follows:(27)$$x_{s,k}^{0}={\hat{x}}_{i,k},P_{s,k}^{0}={\hat{P}}_{1,k}\;x_{s,k}^{i}=P_{s,k}^{i}\left({\left(P_{s,k}^{i-1}\right)}^{-1}x_{s,k}^{i-1}+{\hat{P}}_{i+1,k}^{-1}{\hat{x}}_{i,k}-\Gamma_{i}^{-1}\gamma_{i}\right)\;P_{s,k}^{i}={\left({\left(P_{s,k}^{i-1}\right)}^{-1}+{\hat{P}}_{i+1,k}^{-1}-\Gamma_{i}^{-1}\right)}^{-1}\;\Gamma_{i}=S_{s,k}^{i-1}{\left(D_{s,k}^{i-1}\right)}^{-\frac{1}{2}}S_{i+1,k}D_{\Gamma }S_{i+1,k}^{-1}{\left(D_{s,k}^{i-1}\right)}^{\frac{1}{2}}{\left(S_{s,k}^{i-1}\right)}^{-1}\;\gamma_{i}={\left({\left(P_{s,k}^{i-1}\right)}^{-1}+{\hat{P}}_{i+1,k}^{-1}-2\Gamma_{i}^{-1}+2\eta_{i}I\right)}^{-1}\times \left(\left({\hat{P}}_{i+1,k}^{-1}-\Gamma_{i}^{-1}+\eta_{i}I\right)x_{s,k}^{i-1}+\left({\left(P_{s,k}^{i-1}\right)}^{-1}-\Gamma_{i}^{-1}+\eta_{i}I\right){\hat{x}}_{i+1,k}\right)$$

The mean of the DEI fusion estimation result is ${\hat{x}}_{k}=x_{s,k}^{L-1},$ and the error covariance is ${\hat{P}}_{k}=P_{s,k}^{L-1}$.

In order to visually compare the superiority of EI fusion technique, we analyze the CI fusion, EI fusion technique, and minimization Γ by a simple numerical example. Suppose the two prior estimates x1,x2 obey the Gaussian distribution with a mean of 0 and covariance matrix $P1=\left[\begin{array}{cc} 2 & -1\\ -1 & 1 \end{array}\right]$ and $P2=\left[\begin{array}{cc} 1/3 & 0\\ 0 & 2 \end{array}\right]$, respectively. By converting the Gaussian distribution into a sublevel set for the ellipsoid description, the obtained results are shown in Figure 2. The red curve indicates the ellipsoidal results of CI fusion, the green curve indicates the ellipsoidal results of EI fusion, and the blue zone line indicates the ellipsoid enclosed by the minimization Γ. The results show that the area enclosed by the CI fusion algorithm is larger than the area where the two local estimates intersect, and the estimation results are too conservative, while the EI fusion algorithm is within the area where the two local estimates intersect, and the fusion results are more accurate.

Here, we discuss the consistency of the designed EI fusion estimator, inspired by [15], based on local estimates obeying the Gaussian distribution; we take the results of the fusion of local estimates of sensors 1 and 2 for analysis. The local estimates, after EI fusion, are obtained according to Equation (23) as follows:(28)$${\hat{P}}_{1,k}≼\Gamma_{1},{\hat{P}}_{2,k}≼\Gamma_{1}$$

Taking the inverse of both sides of the symbol simultaneously yields
(29)$$\Gamma_{1}^{-1}≼{\hat{P}}_{1,k}^{-1}$$
which further implies that
(30)$${\hat{P}}_{1,k}^{-1}-\Gamma_{1}^{-1}≽0$$

By adding ${\hat{P}}_{2,k}^{-1}$ to both sides of the symbol simultaneously, we obtain
(31)$${\hat{P}}_{2,k}^{-1}+{\hat{P}}_{1,k}^{-1}-\Gamma_{1}^{-1}≽{\hat{P}}_{2,k}^{-1}$$

According to Equation (25), the first fusion result of $P_{s,k}^{1}≼{\hat{P}}_{2,k}$ and, similarly, $P_{s,k}^{1}≼{\hat{P}}_{1,k}$.

According to (27) and (28), we obtain
$$P_{s,k}^{2}≼P_{s,k}^{1},P_{s,k}^{2}≼{\hat{P}}_{3,k},$$

The following conclusions can be drawn from the collation.
$$P_{s,k}^{2}≼{\hat{P}}_{i,k},\;i=1,2,3,$$

In the *L* − 1 times of the fusion process, based on mathematical induction, it is obtained that
(32)$$P_{s,k}^{L-1}≼{\hat{P}}_{i,k},\;i=1,2,3\cdots ,L$$

In summary, the distributed ellipsoid (DEI) fusion estimator designed in the paper has good consistency, and the fusion estimator outperforms the individual local estimators.

## 5. Numerical Examples

In this section, the proposed DEI fusion is validated by a numerical example in order to intuitively obtain a fusion estimation problem consistent with being able to solve the unknown correlation in a complex multi-sensor system with unknown input disturbances and measurement data transmission delays. First, consider a complex multi-sensor linear time-varying system with unknown external inputs and measurement data transmission delays with the expression:$$x_{k+1}=A_{k}x_{k}+D_{k}d_{k}+\omega_{k}\;y_{i,k}=C_{i,k}x_{k}+v_{i,k}\;,\;i=1,2,3,$$
where the state matrix $A=\left[\begin{array}{ccc} \begin{array}{c} a_{11,k}\\ a_{21,k}\\ a_{31,k} \end{array} & \begin{array}{c} a_{12,k}\\ a_{22,k}\\ a_{32,k} \end{array} & \begin{array}{c} a_{13,k}\\ a_{23,k}\\ a_{33,k} \end{array} \end{array}\right]$.$\;d_{k}$ is a Rayleigh distributed random number obeying parameter 3.

The expression for each element in the state matrix is:$$a_{11,k}=\exp\left[-h+\sin\left(kh\right)-\sin\left(kh-h\right)\right],\;a_{12,k}=0,\;a_{13,k}=0\;a_{21,k}=2\sinh\left(\frac{h}{2}\right)\exp\left[-\frac{3h}{2}+\sin\left(kh\right)-\sin\left(kh-h\right)\right]\;a_{22,k}=\exp\left[-2h+\sin\left(kh\right)-\sin\left(kh-h\right)\right]\;a_{23,k}=0,a_{31,k}=0,a_{32,k}=0\;a_{33,k}=\exp\left[-2h+\sin\left(kh\right)-\sin\left(kh-h\right)\right]$$

The respective measurement matrices of the three sensors are as follows:$$C_{1,k}=\left[\begin{array}{ccc} 1 & \cos\left(\mathrm{kh}\right) & \sin\left(\mathrm{kh}\right) \end{array}\right]\;C_{2,k}=\left[\sin\left(\mathrm{kh}\right)\;\;\;\;2\;\;\;\;\cos\left(\mathrm{kh}\right)\right]\;C_{3,k}=\left[\begin{array}{ccc} \cos\left(kh\right) & \sin\left(kh\right) & 1.5\\ 1 & \sin\left(2kh\right) & \cos\left(2kh\right) \end{array}\right]$$

The unknown input coefficient matrix is $D_{k}={\left[\begin{array}{ccc} 0.1\sin\left(kh\right) & 0.3 & 0.2 \end{array}\right]}^{T}.$ In all formulas, h is 0.2. The covariance matrix of measurement noise is $Q=\mathrm{diag}\left\{1,1,1\right\}$. The process noise in the measurement equations for the three sensors is:$R_{1,k}=0.2$,$\;R_{2,k}=0.3$,$\;\mathrm{and}\;R_{3,k}=\left[0.3\;0.1;0.1\;0.25\right]$. Similar to [26], the time of the measurement data transmission delay is described by a random Poisson distribution with parameters $\lambda_{i}\left(i=1,2,3\right)$, and its probability density function is $f_{i}\left(j\right)$:$$f_{i}\left(j\right)=\frac{\lambda_{i}^{j}e^{-\lambda_{i}}}{j!},j=0,1,\cdots$$

The mean value of the Poisson distribution obeyed by each channel delay time is $\lambda_{1}=5$, $\lambda_{2}=6$, $\lambda_{3}=5$.The buffer length used by each node is $\varepsilon_{1}=\varepsilon_{2}=\varepsilon_{3}=7$. The mean and covariance of the initial state are set as:$${\bar{x}}_{0}={\left[\begin{array}{ccc} 0.1 & 0.1 & 0.1 \end{array}\right]}^{T},{\bar{P}}_{0}=diag\left\{0.1,0.1,0.1\right\}$$

Figure 3 represents the state estimation plots of the DEI fusion estimation for state 1, state 2, and state 3. The black curve indicates the state values without disturbance from the external inputs, the red curve indicates the actual state values of the complex system, and the blue curve indicates the estimated values of the DEI fusion estimator, which shows that the designed DEI fusion estimator can estimate the complex multi-sensor system well.

Based on the property that any Gaussian distribution can be described by the sublevel set ${Ɛ}_{\hat{x},P}=\{x\in R^{n}|{\left(x-\hat{x}\right)}^{T}P^{-1}\left(x-\hat{x}\right)\leq 1\}$, the superiority of the DEI estimator is verified by comparing the area enclosed by the results of the DCI fusion and DEI fusion performances [30].

Figure 4 represents two local estimates ($x_{1},x_{2}$) of the 3D image represented by a sublevel set, and the fusion algorithm estimates the area enclosed by the two ellipsoids. Figure 5 shows the results of the volume of the enclosed region for both fusion algorithms. It can be seen that the conservative estimation of the DCI fusion results in a larger result for the volume of the enclosed region than the DEI fusion result, which validates the superior performance of the DEI fusion estimator.

Next, let us discuss the computational cost of the designed DEI fusion estimator and compare the computational complexity of the centralized estimator with that of the DEI fusion estimator. First, we unify the dimensions of the different measurement equations, i.e., $y_{i,k}\in R^{m_{i}}$. It easily follows that the computational magnitude of the centralized estimator is $O\left({\left(Lm_{i}\right)}^{3}\right)$, and the computational magnitude of the DEI estimator is $O\left(Lm_{i}^{3}\right)$. Since *L* is a positive integer greater than 1, $L<L^{3}$, it can be seen that the computational cost of the DEI estimator algorithm is smaller than that of the centralized algorithm.

In the centralized fusion estimation, the state estimation is handled using the state augmentation method, and the state transfer matrix is an invertible, sparse matrix. According to the nature of a computational complexity analysis, the computational order of magnitude of the centralized fusion estimator is obtained as $O\left(2L^{2}n^{3}+3{\left(Ln\right)}^{2}m_{i}+2Lnm_{i}^{2}+m_{i}^{3}\right)$, while the computational order of magnitude of the designed DEI fusion estimator is $O\left(2n^{3}+3n^{2}m_{i}+5nm_{i}^{2}+m_{i}^{3}+\left(L-1\right)\left(n^{2}m_{i}+nm_{i}\right)\right)$. When $m_{i}=1$, the computational complexity analysis of the designed numerical example shows that the computational order of magnitude of the centralized fusion estimator is $O\left(748\right)$ and that of the DEI fusion estimator is $O\left(148\right)$. It can be seen that the designed DEI fusion estimator significantly reduces the computational cost.

From the analysis of the above results, it can be concluded that the designed DEI fusion estimator solves the problem of low computational cost that centralized fusion does not have and the problem that DCI fusion estimation is too conservative and verifies the superiority of the designed fusion algorithm. Although the distributed fusion algorithm is suboptimal, the proposed DEI fusion estimator with good accuracy and low computational cost is preferred for the estimation of multi-sensor complex systems.

## 6. Conclusions

In this paper, we studied the problem of data fusion estimation for a complex multi-sensor system with two network-induced phenomena of both unknown input disturbances and measurement transmission delays. By treating the unknown input disturbance as a non-informative prior distribution, the measured data transmission delay was represented by a set of independent stochastic Bernoulli processes, and a finite length buffer was added at the link nodes to retrieve the delayed data set. In analyzing the data fusion estimation problem, the MMSE local estimator was designed with a Bayesian framework for a multi-sensor complex system. For the problem of an unknown correlation between local estimates, a DEI fusion estimator that could solve arbitrary correlation was designed, and the consistency of the fusion estimator was demonstrated. In the paper, the superior tracking performance of the designed DEI fusion estimator was analyzed by simulation examples, and the problems of conservative estimation in DCI fusion estimations and high computational costs in centralized fusion were solved. Although information fusion is developing rapidly in this era of rapid development, the research on information fusion estimation needs further efforts.

## Figures and Tables

**Figure 1 sensors-22-04306-f001:**
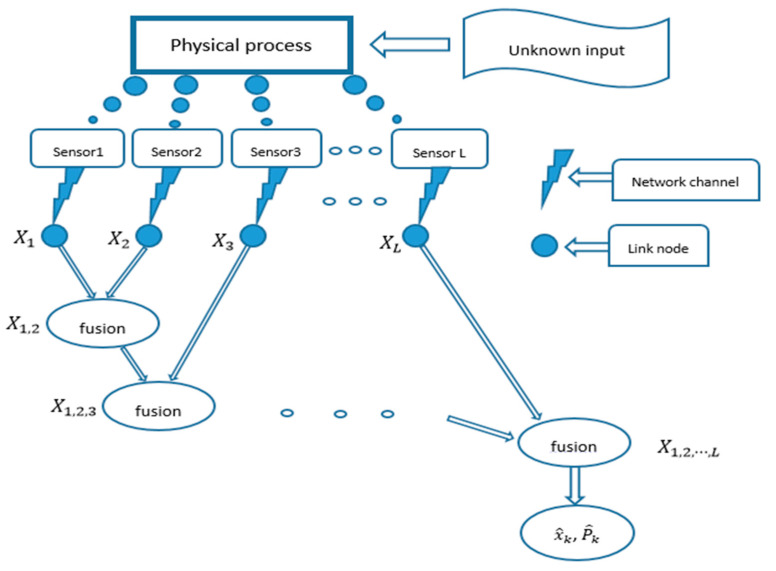
Distributed fusion estimation of complex systems with a multi-sensor.

**Figure 2 sensors-22-04306-f002:**
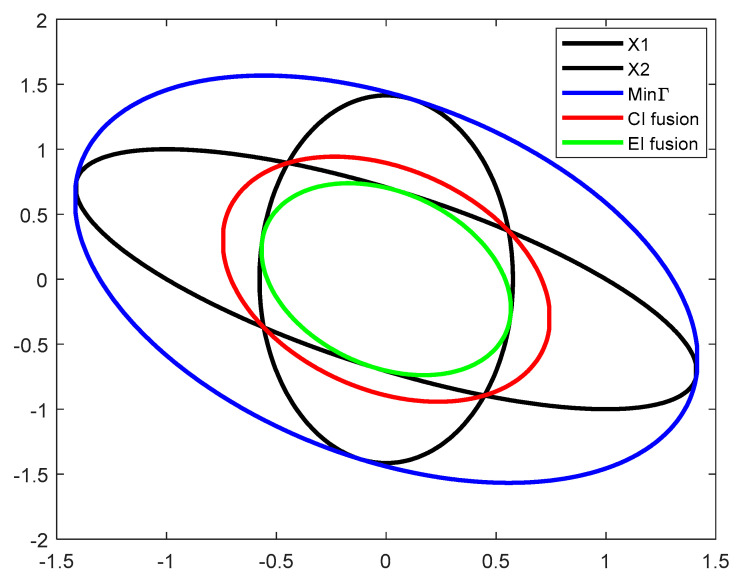
Results of the minimizing Γ, CI fusion, and EI fusion methods for two state ellipsoid estimations.

**Figure 3 sensors-22-04306-f003:**
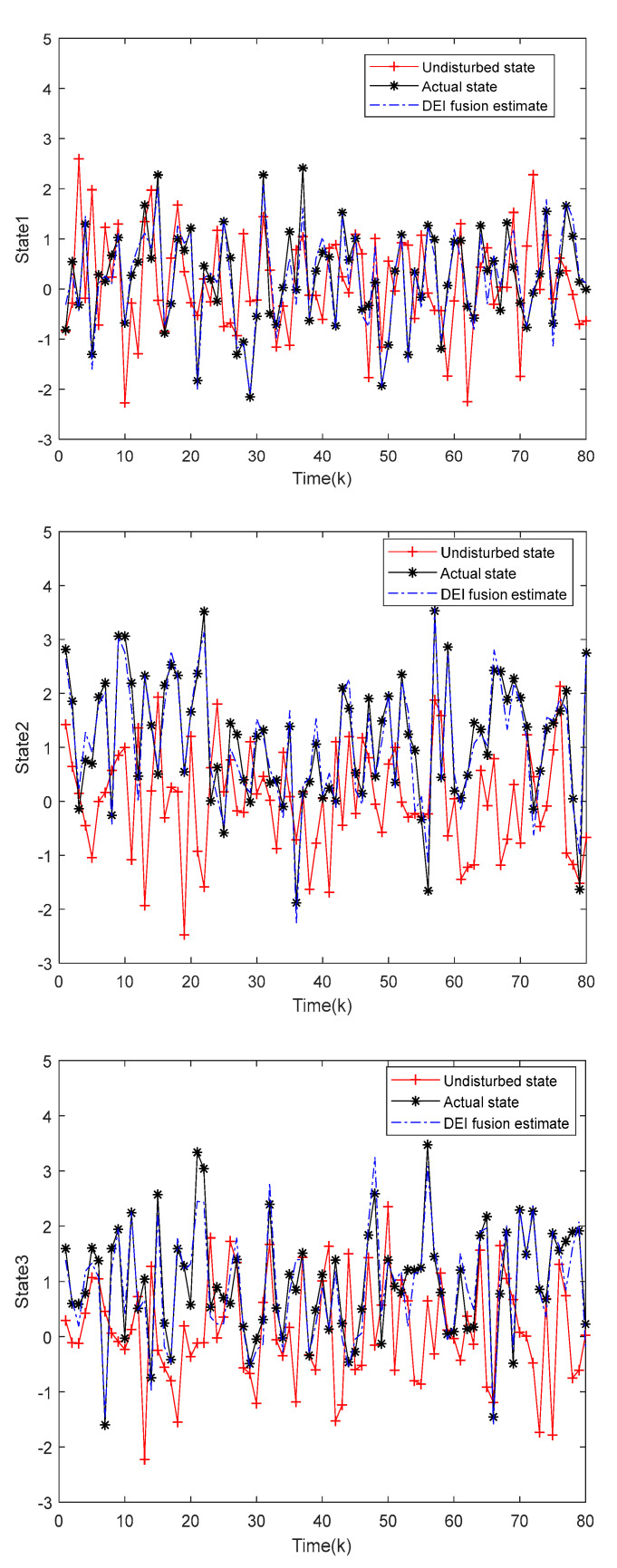
Performance of the Distributed Ellipsoidal Intersection (DEI) fusion estimator in the state estimation.

**Figure 4 sensors-22-04306-f004:**
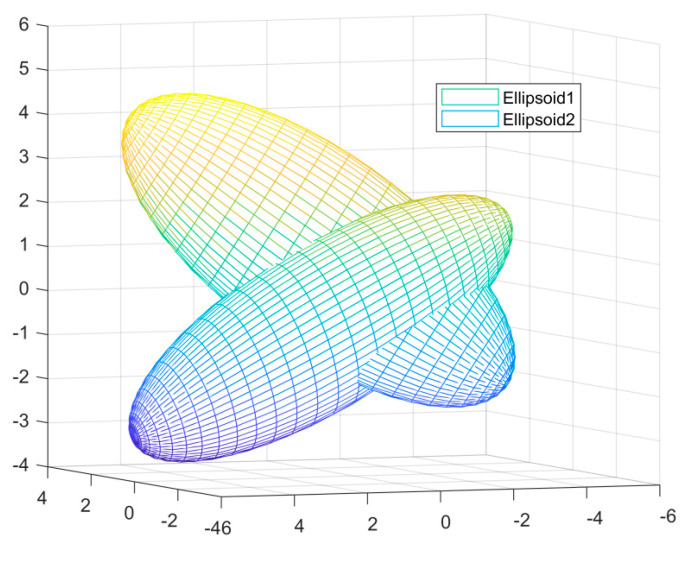
Ellipsoid1 and Ellipsoid2 forms of two locally estimated ($x_{1},x_{2}$) under the sublevel set ${Ɛ}_{\hat{x},P}$.

**Figure 5 sensors-22-04306-f005:**
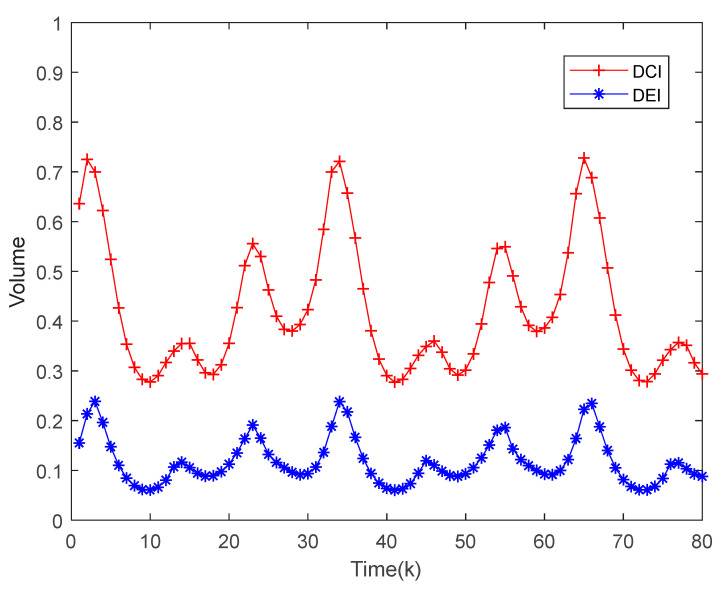
Ellipsoidal volume characterized by the fusion of the DEI and DCI results.

## Data Availability

Not applicable.

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
