# Peer review of "Distributed Ellipsoidal Intersection Fusion Estimation for Multi-Sensor Complex Systems"

_sensors, 2022, doi:10.3390/s22114306_

Round 1

Reviewer 1 Report

The authors focus their study on the problem of distributed ellipsoidal intersection fusion estimation for linear time varying multisensor complex systems while considering a known input disturbances and measurement data transmission delays. The manuscript is overall well written and easy to follow and the authors have well thought out their main contributions. The provided theoretical analysis is concrete, complete, and correct and the authors have provided all the intermediate steps in order to enable the reader to easily follow it. The provided numerical results are also rich in order to show the pure operation and the performance of the proposed framework. The authors are encouraged to consider the following suggestions provided by the reviewer in order to improve the scientific depth of their manuscript, as well as they need to address the following comments in order to improve the quality of presentation of their manuscript. Initially, in Section 1, the authors need to substantially revise the provided related work and elaborate more on the research contributions that have already been performed in the literature and the research gap that the authors try to address. The  currently provided related work needs to discuss several existing methods that have been used in the literature in order to improve the operation of sensors systems, such as Interest, energy and physical-aware coalition formation and resource allocation in smart IoT applications, doi: 10.1109/CISS.2017.7926111, and group the research works in a meaningful manner and then focus on the specific problem examined in the paper. Furthermore, the authors need to include an additional subsection within section 5, in order to provide the theoretical analysis of the computational complexity of the proposed framework. Based on the previous comment, the authors need to include some additional numerical results quantifying the computational complexity of the proposed method. Furthermore, in Section 5, the authors need to provide some comparative results to the state of the art in order to quantify the drawbacks and benefits of the proposed approach. Finally, the overall manuscript needs to be checked for typos, syntax, and grammar errors in order to improve the quality of its presentation.

Author Response

很荣幸您能成为我论文的审稿人,你们努力工作!我已经回复了您在附件中所有有价值的意见。如果有任何更正,请帮助我指出来。谢谢!

Reviewer 2 Report

  1. Please add a space between the previous word and citation on the first page.
  2. In the last paragraph in section 1, please highlight the proposed novelty.
  3. Please provide a manuscript outline at the end of Section 1 to provide a connection between the sections.
  4. vector??appear”, “model??using” - above eq. (3).
  5. Other typos in the first paragraph after eq. (3).
  6. Eq. (5): please correctly write the equation.
  7. Eq. (6) looks strangely positioned.
  8. Section 3, small typos when writing the equations.
  9. Punctuation before Eq. (7). Moreover, add the corresponding punctuation after each equation.
  10. Quite a lot of typos in the entire manuscript. Please pay more attention to details.
  11. Comparison with state-of-the-art?
  12. Please provide a motivation for employing the proposed method.
  13. Figure 4 legend is unclear.

The literature review was performed quite well.

The method is described in detail, however, please highlight the proposed novelties and motivation.

Please check the manuscript for typos.

Author Response

(The authors gave the same response as above.)

Round 2

Reviewer 1 Report

The authors have addressed the reviewers comments.